# Evaluating the Evidence in Clinical Studies of Vitamin D in COVID-19

**DOI:** 10.3390/nu14030464

**Published:** 2022-01-21

**Authors:** Tom D. Thacher

**Affiliations:** Department of Family Medicine, Mayo Clinic 200 First Street SW, Rochester, MN 55905, USA; Thacher.Thomas@Mayo.edu

**Keywords:** nutrition, cholecalciferol, 25-hydroxyvitamin D, infection, study design, epidemiology, clinical trials

## Abstract

Laboratory evidence provides a biological rationale for the benefits of vitamin D in COVID-19, and vitamin D supplementation is associated with reduced risk of respiratory infections. Most of the clinical studies of vitamin D in COVID-19 have been observational, and the most serious problem with observational study design is that of confounding. Observational studies typically assess the relationship of 25(OH)D values with COVID-19 outcomes. Many conditions associated with low vitamin D status are also associated with worse COVID-19 outcomes. Randomized controlled trials (RCTs) overcome the problem of confounding, typically comparing outcomes between groups receiving vitamin D supplementation or placebo. However, any benefit of vitamin D in COVID-19 may be related to the dose, duration, daily vs. bolus administration, interaction with other treatments, and timing of administration prior to or during the illness. Serum 25(OH)D values >50 nmol/L have been associated with reduced infection rates, severity of COVID-19, and mortality in observational studies. Few RCTs of vitamin D supplementation have been completed, and they have shown no benefit of vitamin D in hospitalized patients. Vitamin D may benefit those with mild or asymptomatic COVID-19, and those with greater 25(OH)D values may have lower risk of acquiring infection. Because those at greatest risk of COVID-19 are also at greatest risk of vitamin D deficiency, it is reasonable to recommend vitamin D supplementation 15–20 mcg (600–800 IU) daily for the general population during the COVID-19 pandemic. Vitamin D doses greater than 100 mcg (4000 IU) daily should not be used without monitoring serum 25(OH)D and calcium.

## 1. Theoretical Benefits of Vitamin D in COVID-19

Vitamin D deficiency has classically been associated with the bone diseases of rickets in growing children and osteomalacia in adults [1]. Increasing interest in the non-skeletal effects of vitamin D relates to the finding of vitamin D receptors (VDR) widely distributed in most human tissues, as is the 1α-hydroxylase enzyme (CYP27B1). Within these tissues, 1α-hydroxylase converts 25(OH)D to 1,25(OH)_2_D in order to exert localized effects (paracrine effects), without altering serum 1,25(OH)_2_D concentrations. Local 1,25(OH)_2_D enters the cell nucleus to influence the expression of genes, unrelated to calcium absorption or bone metabolism.

Severe COVID-19 results from an exuberant and dysregulated immune response to the SARS-CoV-2 virus. The term COVID-associated acute respiratory distress syndrome (CARDS) has been used to describe the similar clinical manifestations and pathophysiology of severe COVID-19 and those of acute respiratory distress syndrome, including multisystemic effects from the release of proinflammatory cytokines. 

The COVID-19 pandemic continues, and current evidence for the role of vitamin D in the treatment and prevention of COVID-19 deserves regular review. Studies of vitamin D in COVID-19 are based on a biological rationale for the benefits of vitamin D in COVID-19. There are three primary lung defenses against infection: airway epithelia, alveolar macrophages, and dendritic cells involved in cytokine production. All these cells express 1a-hydroxylase and are capable of locally producing 1,25(OH)_2_D, which acts as an important immune and inflammatory modulator. Vitamin D can alter the expression of genes involved in infection and inflammation and could theoretically decrease the severity of COVID-19 infection. Specifically, vitamin D could have benefit in COVID-19 in three different ways: (1) reducing the risk of acquiring SARS-CoV-2 infection, (2) enhancing viral neutralization and clearance, and (3) reducing the severity of the inflammatory response [2]. 

Laboratory evidence demonstrates that 1,25(OH)_2_D promotes the expression of antimicrobial proteins cathelicidin and β-defensin2 by pulmonary macrophages and epithelium. It also suppresses antigen presentation by dendritic cells and activation of T cells, thereby inhibiting proinflammatory cytokine production. Additionally, 1,25(OH)2D regulates ACE2 to reduce vascular sensitivity and SARS-CoV-2 attachment to ACE2 receptors [2].

Daily or weekly doses of vitamin D, but not large bolus doses, were associated with a reduced risk of respiratory infection in a meta-analysis of randomized controlled trials of vitamin D supplementation, particularly in subjects with 25(OH)D values <25 nmol/L [3]. A single high dose of vitamin D can induce the 24-hydroxylase enzyme (CYP24A1), resulting in greater production of 24,25(OH)_2_D relative to 25(OH)D than daily supplementation [4]. A daily dose of vitamin D may increase 25(OH)D with less diversion of 25(OH)D to 24,25(OH)_2_D than bolus dosing.

## 2. Strengths and Limitations of Clinical Study Designs

The aim of this narrative review is to equip the reader with a framework to evaluate the evidence related to vitamin D in the treatment and prevention of COVID-19. Studies have been selected from a search of English language, peer-reviewed publications on PubMed, using the terms “vitamin D” and “COVID”. Selected studies represent those with larger sample sizes and higher study quality, including meta-analyses. The GRADE scoring methodology can be used as a means of formally evaluating study quality and making recommendations [5], and the focus of this review is to enable the reader to understand the limitations of study design and appropriately assess study quality. Many trials are yet to be completed: clinicaltrials.gov on 25 January 2022 displayed 67 active trials of vitamin D in COVID-19. 

Two broad categories of study design are used in most clinical studies: observational and experimental studies. Observational study designs include case-control, cross-sectional, and cohort studies. In observational studies of vitamin D, the serum 25(OH)D concentration, reflecting vitamin D status, is typically the independent variable. Observational studies can only demonstrate associations and not prove causation. Experimental studies are generally designed as randomized controlled trials (RCT). Vitamin D supplement intake is typically the independent variable. Results of RCTs are usually considered higher quality evidence than those of observational designs.

The most serious problem with observational studies is that of confounding. A confounder is a disease or behavior that is both associated with 25(OH)D and a risk factor for the outcome. It is possible to statistically adjust for confounding variables if they have been measured and are included in multivariable models. High risk conditions associated with severe COVID-19 include age >65 years, obesity, chronic kidney disease, diabetes, cerebrovascular disease, heart conditions, chronic lung diseases, chronic liver disease, cancer, mental health disorders, pregnancy, and smoking [6]. All increase the risk of adverse outcomes from COVID-19, but most of these conditions have been associated with lower vitamin D status, as measured by serum 25(OH)D. In any study of COVID outcomes, these should be considered confounding variables, because they are related both to vitamin D status and to COVID-19 outcomes. Studies must report and adjust for these confounding variables in analysis of outcomes. Sample sizes should be large enough to allow for adjustment, i.e., generally at least 10 subjects for each variable in an adjusted multivariate model.

Multiple examples of confounding in observational studies of vitamin D are applicable in studies of vitamin D in COVID-19. Obesity is both associated with lower 25(OH)D and worse outcomes in COVID-19. Seasonal variation of 25(OH)D will be associated with diseases having seasonal variation, like respiratory illnesses. Additionally, seasonal variation in gene expression related to immune responses has been described [7], and the benefits of vitamin D on health outcomes have been proposed to be season dependent [8]. People with chronic illnesses have less outdoor activity and lower 25(OH)D levels. Chronic disease and critical illness can lower 25(OH)D, so low 25(OH)D can be the result rather than the cause of more severe illness. Serum 25(OH)D values may be inversely related to inflammatory and acute phase markers in severe illness [9,10]. Dietary intake of foods with vitamin D may improve vitamin D status, but other nutrients in those foods may be related to COVID-19 outcomes. People from racial groups with dark skin generally have lower serum 25(OH)D concentrations than white Caucasians and are at greater risk for severe COVID-19. 

Besides vitamin D intake and sunlight exposure, additional factors influence 25(OH)D values. The methodology for measurement of 25(OH)D can be an important source of variation between studies, particularly when cut-point values are used, highlighting the need for standardized 25(OH)D measurements [11]. Polymorphisms of genes related to vitamin D transport and metabolism are predictive of 25(OH)D concentrations [12]. The 25(OH)D concentration indicative of adequate vitamin D status and the effect of 1,25(OH)_2_D on genetic expression is likely to be highly variable between individuals [13]. Serum 25(OH)D is a marker of vitamin D supply but not a functional measure of the action of 1,25(OH)_2_D on vitamin D receptor-mediated gene expression. The degree of vitamin D deficiency that has detrimental effects on inflammatory and immune regulation is unknown. Investigators identified two susceptibility loci with genome-wide significance for severe COVID-19 with respiratory failure in Spain [14], and low 25(OH)D levels could theoretically be linked to genetic loci also associated with an increased risk of severe COVID-19. It is the free 25(OH)D, unbound to vitamin D binding protein (DBP), that is available intracellularly. DBP affects the bioavailability of 25(OH)D to monocytes, and immune responses may be related to DBP polymorphisms [2,15].

Limitations of the current evidence base need to be recognized. Many of the studies assessing the relationship of vitamin D status with clinical outcomes are observational, retrospective studies. Low vitamin D status could be a cause or consequence of severe illness. Serum 25(OH)D values may be inversely related to inflammatory and acute phase markers in COVID-19 [16,17]. Low serum 25(OH)D levels may be a marker of chronic illness or mortality risk and may not necessarily indicate a therapeutic benefit of improving the vitamin D status. For example, patients with chronic illnesses may have low vitamin D status resulting from their disease, inadequate dietary intake, or limited sun exposure, but it is their chronic disease that puts them at greater risk of severe COVID-19. 

There are additional limitations of observational studies. One is selection bias: patients who had 25(OH)D measurements available are those selected for study, and these subjects may differ from those who did not have 25(OH)D measured. Healthy user bias may result when healthier people are more likely to take vitamin D than less healthy individuals. One major issue with observational studies, given their retrospective nature, results from post-hoc analysis. Investigators do not always provide a pre-specified hypothesis or specify the primary outcome tested in observational studies. A pre-specified hypothesis is needed to correctly apply significance testing with a *p* value of 0.05. Multiple post-hoc comparisons may be performed examining multiple outcomes, varying subgroups of patients, or alternative cut-points for 25(OH)D. Each statistical comparison increases the risk of a type 1 statistical error, i.e., concluding that a relationship with 25(OH)D is significant when it is not. For example, if one makes 20 statistical comparisons, the probability of finding one comparison with *p* < 0.05 is not 5%, but 64%. For these reasons, a post-hoc analysis should be considered exploratory, to identify possible relationships that need confirmation.

Other limitations are related to reporting study outcomes, and these are more likely to occur with observational studies than with RCTs. The first of these is publication and reporting bias. Medical journals are more likely to publish studies that show potential benefit of an intervention than studies with negative results. This can skew the published literature toward positive results. Negative results are frequently due to inadequate study power, and authors may abandon efforts to publish negative studies. The second concern relates to preprint servers. These allow authors to post their manuscript on a public server prior to peer-review. This is intended to allow authors to quickly disseminate research findings, and the use of preprint servers has proliferated during the COVID-19 pandemic. Because these manuscripts are not peer-reviewed, it is difficult to be assured of the quality of the work and more likely for fraudulent material to be published than in peer-reviewed publications. Their results are intended to be considered preliminary.

Compared with observational studies, a much stronger level of evidence is provided by well-designed RCTs. The major advantage of RCTs is the control for confounding variables, even those that are not measured, because the randomization process generally balances confounding variables between vitamin D and placebo groups. RCTs have not consistently confirmed beneficial effects of vitamin D found in observational studies. Although RCTs generally represent a higher quality of evidence than observational studies, they also have their limitations. Subjects in the control group may also take vitamin D through diet or supplements, potentially attenuating any observed benefit. RCTs may have an inadequate number of persons with vitamin D deficiency, or subjects with vitamin D deficiency may have been excluded from the RCT. Additional vitamin D may be of no benefit for persons with an adequate vitamin D status. The dose, duration, or bolus vs. daily administration of vitamin D may be related to benefit [18]. Finally, the timing of vitamin D administration in relation to illness onset is essential to consider in COVID-19, because the beneficial effects may vary by the stage or severity of illness.

The temporal course of COVID-19 has been described in three phases. During the first week, there is viral replication of SARS-CoV-2. Severe COVID-19 typically develops in the second week, in up to 5% of infected patients. Severe COVID-19 results from exuberant and dysregulated immune responses to high viral loads. The term COVID-associated acute respiratory distress syndrome (CARDS) has been used to describe the similar clinical manifestations and pathophysiology of severe COVID-19 to acute respiratory distress syndrome, including multisystemic effects from the release of proinflammatory cytokines, also called a cytokine storm. Post-acute sequelae of SARS-CoV-2 (PASC or long COVID) refer to persistent symptoms after recovery, such as chronic fatigue, headache, brain fog, and dizziness. Any benefit of vitamin D in COVID-19 may vary with the phase or severity of illness. This is not a unique consideration for vitamin D. For example, dexamethasone is beneficial to those hospitalized with COVID-19 on oxygen but not in those who are not on oxygen. Some monoclonal antibody treatments for COVID-19 are beneficial in the first phase, but not in the second phase. Tocilizumab is beneficial in hospitalized patients on corticosteroids, but not when used alone. Vitamin D may be beneficial for a specific phase or severity of illness, or when used in combination with other treatments. Table 1 summarizes selected limitations to be considered related to study design in vitamin D and infection.

## 3. Observational Studies

Studies of the association of vitamin D status with the severity of COVID-19 have been carried out in many countries. The majority of studies include patients with mean ages in the range of 50–65 years, and the age-related increase in risk begins at age 50 and rises continuously with advancing age. Among 216 hospitalized patients in Spain, 82% had 25(OH)D levels below 50 nmol/L, compared with 47% in population-based controls, matched only for sex [16]. However, there was no relationship between the severity of COVID-19 with 25(OH)D level. In the same study, lower levels of 25(OH)D were associated with higher levels of ferritin, D-dimer, and CRP, which are inflammatory markers that are commonly elevated in patients with COVID-19. This suggests that low vitamin D status may also be a marker of more severe inflammation.

Using a retrospective cohort design in racially diverse, hospitalized patients with COVID-19 in the U.S., investigators found that patients with serum 25(OH)D values <75 nmol/L within 6 months before or during hospitalization for COVID-19 had increased mortality and need for invasive mechanical ventilation [19]. Among hospitalized patients in the UAE with COVID-19 and 25(OH)D measured on admission, a 25(OH)D concentration <30 nmol/L was significantly associated with 1.8 times greater odds of severe illness and 2.6 times greater odds of death [20]. In a study of over 80,000 patients in the UK with COVID-19 and a 25(OH)D measured within 12 months prior to diagnosis, a value of 25(OH)D <50 nmol/L was associated with 2.4 times greater odds of hospitalization [21]. Among hospitalized patients with COVID-19 in Spain and 25(OH)D measured on admission, a 25(OH)D value <50 nmol/L was associated with 4.2 times greater odds of admission to ICU but not with mortality [22].

A study of hospitalized patients with COVID-19 in the US and 25(OH)D measured during the prior year explored multiple outcomes and included adjustment for multiple confounding variables [23]. The subgroup of patients 65 years-old or greater with 25(OH)D values >75 nmol/L had reduced odds of death, ARDS, and severe sepsis, compared with those having 25(OH)D ≤75 nmol/L, possibly indicating a higher inflammatory burden of COVID-19 in older patients.

In a very large population-based study in Spain, subjects who were on a vitamin D supplement were compared with propensity-matched controls that were not taking vitamin D [24]. Patients taking vitamin D had a slightly lower risk of SARS-CoV2 infection (4.0% vs. 4.2%). In a subgroup analysis, those taking vitamin D with 25(OH)D values >75 nmol/L had a lower risk of infection (3.3%) and mortality (0.6%) than those not taking vitamin D with 25(OH)D values <50 nmol/L (5.6% and 1.3%, respectively).

However, not all observational studies have found a relationship between vitamin D status and outcomes of COVID-19. Studies in the USA [25,26], Italy [27], the UK [28], and India [29] failed to confirm a beneficial effect of greater 25(OH)D levels with outcomes of length of stay, days on oxygen, ICU admission, need for assisted ventilation, or mortality, but some of these studies may have lacked sufficient power to detect a relationship between vitamin D status and less frequent outcomes, like mortality.

Vitamin D status may interact with other treatments for COVID-19. Investigators in the UK compared outcomes associated with vitamin D status before (March–April 2020) and after (September–December 2020) the use of dexamethasone in hospitalized patients [30]. Vitamin D deficiency was associated with elevated CRP and need for ventilation in hospitalized COVID-19 patients prior to use of dexamethasone but not during the interval of dexamethasone use. The primary outcome was mortality, and no mortality difference was evident during either interval. Differences in ventilation and mortality rates between these two time intervals suggested greater severity of illness in the dexamethasone-treated group. Dexamethasone may attenuate the adverse effects of vitamin D deficiency. Glucocorticoids increase 24-hydroxylase gene (*CYP24A1*) transcription [31] and are associated with lower 25(OH)D concentrations, but they also increase *VDR* transcription which can enhance 1,25(OH)_2_D effects [32].

A meta-analysis combined the results of observational studies in 2020 [33]. Vitamin D deficiency was variably defined as total 25(OH)D) level less than 30 nmol/L (seven studies), less than 50 nmol/L (eight studies) and less than 62.5 nmol/L (one study). Vitamin D deficiency was associated with 2.5 times greater odds of mortality. Vitamin D deficiency was also associated with higher rates of hospital admission and longer hospital stays but no significant difference in ICU admissions. However, the authors found substantial heterogeneity and a high risk of bias in the included studies due to a lack of control for confounding variables.

A retrospective analysis of 191,779 individuals tested for SARS-CoV-2 at a national reference laboratory and matched with 25(OH)D results in the preceding 12 month found lower SARS-CoV-2 positivity rates among those with greater 25(OH)D concentrations [34]. This relationship was consistent irrespective of latitude, race, age, and sex. Similarly, in a retrospective population study of 7807 patients in Israel tested for SARS-CoV-2 with any previous 25(OH)D level, those with 25(OH)D <50 nmoL/L had a 60% greater odds of testing positive for COVID-19, while adjusting for the fact that those who had positive tests were significantly younger than those who tested negative (36 vs. 47 years) [35].

## 4. Randomized Controlled Trials

RCTs of vitamin D supplementation in COVID-19 are necessary to conclusively demonstrate benefit [36]. A RCT comparing a single bolus dose of vitamin D 540,000 IU with placebo in critically ill patients (unrelated to COVID-19) found no benefit in patients with baseline 25(OH)D levels less than 50 nmol/L [37]. Subjects in the vitamin D group had no survival benefit, even among those with severe vitamin D deficiency (25(OH)D <30 nmol/L) and, relevant to COVID-19, mortality was greater in subgroups with infection or ARDS who received vitamin D.

A methodologically sound RCT of vitamin D in COVID-19 was performed in 240 hospitalized patients in Brazil, randomized to a single oral dose of vitamin D 200,000 IU or placebo [38]. The study excluded those with severe illness, admitted to ICU, or requiring invasive ventilation. The primary outcome was the probability of hospital discharge over the course of the hospital stay. The investigators found no difference in the hospital length of stay between the vitamin D and placebo groups, even in a post hoc analysis of those with baseline 25D <50 nmol/L. They found no difference between groups in secondary outcomes of mortality, ICU admission, or mechanical ventilation. On limitation of this study was that the mean time from the onset of symptoms to randomization was 10.3 days and from hospitalization to randomization was 1.4 days. Giving vitamin D at the time of hospitalization may be too late to observe benefit against the severity of illness.

A meta-analysis combining the results of three RCTs and two quasi-experimental studies of vitamin D in COVID-19 found no conclusive evidence that vitamin D supplementation reduced mortality, invasive ventilation, or ICU admission [39]. A RCT in 76 patients in Spain that was included in the meta-analysis used oral calcifediol (25(OH)D) rather than vitamin D and found that calcifediol significantly reduced the risk of ICU admission (adjusted OR 0.03) [40].

Vitamin D may have greater benefit if given earlier in COVID-19 or prior to infection, similar to monoclonal antibodies that have benefit early in COVID-19 to prevent severe illness and hospitalization. In a RCT of vitamin D 60,000 IU/day for 7 days in 40 subjects in north India with baseline 25(OH)D <50 nmol/L and mild or asymptomatic COVID-19, viral clearance before day 21 occurred three times more frequently in those randomized to vitamin D (63% vs. 21%) [41]. 

## 5. Conclusions

To summarize the evidence, a serum 25(OH)D >50 nmol/L has been associated with reduced infection rates, reduced severity of COVID-19, and reduced mortality in observational studies, but observational studies have a high risk of bias and are limited by the relationship of vitamin D status with comorbidities. Few RCTs of vitamin D supplementation have been completed, and they have shown no benefit of vitamin D in hospitalized patients. A small study suggested benefit of vitamin D in mild or asymptomatic COVID-19. Those with greater 25(OH)D levels may have lower risk of acquiring infection. The timing of vitamin D administration and phase of illness may be critical to observe benefit in COVID-19. Randomized controlled trials are necessary to confirm beneficial effects of vitamin D suggested by observational studies. Further studies of the dose, timing, and interaction of vitamin D with other treatments are indicated. 

Because those at greatest risk of COVID-19 are also at greatest risk of vitamin D deficiency, it is reasonable to recommend vitamin D supplementation for the general population during the COVID-19 pandemic. The recommended dietary allowances (RDA) for vitamin D in the US were set by the US National Academy of Medicine to achieve a concentration of 25(OH)D of 50 nmol/L [42]. The RDA was set at 15 mcg (600 IU) daily for persons aged 1–70 years and 20 mcg (800 IU) daily for those over age 70 years. An upper limit vitamin D intake of 100 mcg (4000 IU) daily does not require monitoring, but higher intakes should be monitored. No harms are associated with this dose range of vitamin D, and there is potential benefit in reducing the severity of COVID-19 and risk of infection. Doses of vitamin D greater than 100 mcg (4000 IU) daily have no established role in the treatment or prevention of COVID-19, and excessive vitamin D can lead to toxicity, manifested as hypercalcemia and nephro-calcinosis.

## Figures and Tables

**Table 1 nutrients-14-00464-t001:** Selected Limitations to be Considered Related to Study Design in Vitamin D and Infection.

**Observational Studies**
Independent variable: vitamin D status (serum 25(OH)D concentration)
Confounding variables: associated with both 25(OH)D and the outcome
Sample size must be adequate to adjust for known confounding variables
Seasonal variation of 25(OH)D and respiratory illnesses
Those with chronic illness have less sunlight exposure to produce 25(OH)D
Obesity is associated with both lower 25(OH)D and adverse outcomes
25(OH)D may be inversely related to inflammatory markers in severe illness
Racial groups with dark skin may have lower 25(OH)D and different outcomes than Caucasian whites
Vitamin D fortified foods increase 25(OH)D, but other nutrients in food may be related to outcomes
25(OH)D level is related to genes involved in vitamin D transport and metabolism, which could be linked to other genes affecting disease outcomes
Laboratory variation in 25(OH)D measurements and methodology requires standardization
Selection bias: Those with 25(OH)D measurements available were selected for study. They likely differ from those who did not have 25(OH)D measured.
Healthy user bias: Those who take vitamin D may be healthier than those who do not.
Post-hoc analysis: A pre-specified hypothesis is needed to correctly apply significance testing. Analyses of multiple outcomes, subgroups, and 25(OH)D cut points can lead to erroneous conclusions (statistical type 1 error).
Publication and reporting bias: Journals are more likely to publish studies that show potential benefit of an intervention than studies with negative results.
Preprint server publications are not peer-reviewed and results should be considered preliminary.
**Randomized Controlled Trials**
Independent variable: vitamin D supplementation (dose of vitamin D)
Control group may also take vitamin D, potentially attenuating any observed benefit
Inadequate number of persons with vitamin D deficiency (or vitamin D deficient subjects excluded)
Dose and duration of vitamin D may be related to benefit
Formulation of vitamin D may be related to benefit (e.g., cholecalciferol vs. calcifediol)
Daily vs. bolus dosing of vitamin D may have different metabolic effects
Timing of vitamin D administration in relation to illness onset, stage of disease, or illness severity
Interaction of vitamin D with other treatments for disease (e.g., corticosteroids)

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
