# Peer review of "Evaluating the Evidence in Clinical Studies of Vitamin D in COVID-19"

_nutrients, 2022, doi:10.3390/nu14030464_

Round 1

Reviewer 1 Report

The author gives a narrative review on the evidence of clinical studies of vitamin D in COVID 19. He explains the problems and possibilities of bias in observational studies and reports and summarizes the relevant clinical studies.  He also refers the existing randomized clinical trials. The paper appears appropriate to inform readers not specialized in the field on the current state of evidence.

Specific comments: 

Chapter 2:  grammar should be corrected :

Other limitations are related to the reporting of study outcomes, and these are more are more likely to occur with observational studies than RCTs. The first of these is publicatio and reporting bias. Medical journals are more likely to publish studies with that show potential benefit of an intervention than studies with negative results. This can skew

4. Randomized Controlled Trials:   The author cites the meta-analysis of  Rawat  et al. , which reports randomized placebo controlled trials coming to the conclusion that there is no significant effect of vitamin D treatment. This meta-analysis includes the work of Castillo et al.    The paper of Castillo et al. is therefore not especially mentioned by the author of the present review. Because the Castillo paper ist the only one demonstrating a therapeutic/preventive  effect of vitamin D (reducing admission to ICU and mechanical ventilation), it should be mentioned and reported briefly. 

E. Castillo et al. J Steroid Biochem Mol Biol 2020;203:105751

Reviewer 2 Report

This is a narrative review regarding vitamin D status and possible protective benefits from acquiring or recovery from Covid-19.  Generally the review is balanced and touches upon important clarifications that are needed in the field.  To date there is no strong evidence that vitamin D is protective against Covid-19.

Comments to enhance the review:

  • During infection, the mechanism by which serum 25(OH)D declines should be discussed so that the readers understand why this is a major confounding factor. Discuss how VDBP is part of the immune response as is albumin and that the negative acute phase response is in part responsible for lower serum 25(OH)D in illness.
  • The conclusion sections of the abstract and paper that refer to recommendations for the general population cite a single paper (ref 31) and one that is not for the general population. The general population is not a clinical population and thus the recommendations from the National Academy of Medicine (formerly Institute of Medicine) should be used.  The recommendation have not changed during the pandemic.  Generally 400 IU per day is enough for most of the population, and 600 to 800 IU/d depending on age is recommended for individuals to help achieve and maintain a serum 25(OH)D of 50 nmol/L and above.  The tolerable upper level of 4000 IU/d should be mentioned as a population level that does not require monitoring, however higher intakes should be monitored.  The maximum amount of vitamin D in supplements (over the counter) are also not mentioned.
  • The age groups most at risk are not addressed well in the review vs those studied.
  • The quality of the observational and RCT should be included using an accepted scoring system.
  • The metabolism of bolus dose is not adequately discussed in comparison to weekly or daily; for the hospitalized patient bolus has advantages but also drawbacks.
  • There is some mention of dexamethasone, it also alters vitamin D metabolism (VDBP and CYPs) and this should be discussed.
  • A table of the studies reviewed should be included.
  • A table of trials yet to be completed could also enhance the review.

Reviewer 3 Report

The paper deal with an interesting topic. However, major limitations are present and should be addressed:

- A brief introduction to explain the need of this paper is essential

- The aims of this paper should be clearly reported

- a more systematic analysis of the literature should be performed. How did the author selected the papers included in its paper?

- some important papers are missed on the topic (e.g. PMID: 33925932)

- some recent evidence suggest that the seasonality is important not only for the levels of vitamin D but also to determine its biological activity (PMID: 33420273, PMID: 25965853). The authors should discuss this new interesting topic

- the author fail to address the role of the technique used to assess vitamin D that it is crucial to evaluate the results of the studies

Round 2

Reviewer 3 Report

Thank you for addressing my concerns